# Circular RNAs: A New Approach to Multiple Sclerosis

**DOI:** 10.3390/biomedicines11112883

**Published:** 2023-10-24

**Authors:** Raffaele Sciaccotta, Giuseppe Murdaca, Santino Caserta, Vincenzo Rizzo, Sebastiano Gangemi, Alessandro Allegra

**Affiliations:** 1Hematology Unit, Department of Human Pathology in Adulthood and Childhood “Gaetano Barresi”, University of Messina, Via Consolare Valeria, 98125 Messina, Italy; sciaccottaraffaele@gmail.com (R.S.); santino.caserta@polime.it (S.C.); aallegra@unime.it (A.A.); 2Department of Internal Medicine, University of Genova, Viale Benedetto XV, 16132 Genova, Italy; 3IRCCS Ospedale Policlinico S. Martino, 16132 Genova, Italy; 4Department of Clinical and Experimental Medicine, University of Messina, Via Consolare Valeria, 98125 Messina, Italy; vincenzo.rizzo@unime.it; 5Allergy and Clinical Immunology Unit, Department of Clinical and Experimental Medicine, University of Messina, Via Consolare Valeria, 98125 Messina, Italy; gangemis@unime.it

**Keywords:** circRNA, multiple sclerosis, non-coding RNA, epigenetic, immune system

## Abstract

Multiple sclerosis, a condition characterised by demyelination and axonal damage in the central nervous system, is due to autoreactive immune cells that recognise myelin antigens. Alteration of the immune balance can promote the onset of immune deficiencies, loss of immunosurveillance, and/or development of autoimmune disorders such as MS. Numerous enzymes, transcription factors, signal transducers, and membrane proteins contribute to the control of immune system activity. The “transcriptional machine” of eukaryotic cells is a complex system composed not only of mRNA but also of non-coding elements grouped together in the set of non-coding RNAs. Recent studies demonstrate that ncRNAs play a crucial role in numerous cellular functions, gene expression, and the pathogenesis of many immune disorders. The main purpose of this review is to investigate the role of circular RNAs, a previously unknown class of non-coding RNAs, in MS’s pathogenesis. CircRNAs influence post-transcriptional control, expression, and functionality of a microRNA and epigenetic factors, promoting the development of typical MS abnormalities such as neuroinflammation, damage to neuronal cells, and microglial dysfunction. The increase in our knowledge of the role of circRNAs in multiple sclerosis could, in the future, modify the common diagnostic–therapeutic criteria, paving the way to a new vision of this neuroimmune pathology.

## 1. General Considerations on Multiple Sclerosis

The most common autoimmune condition that results in non-traumatic neurological impairment in young people is multiple sclerosis (MS) [1]. MS has the highest prevalence in North America, Western Europe, and Australasia (>100 cases per 100,000 people) and lowest in countries close to the equator (30 cases per 100,000 people) [2]. Females are three times as likely as males to develop the condition, and although MS can manifest at any age, the average occurrence is between twenty and forty years old, with about ten percent of the cases starting under the age of eighteen [3]. Cognitive dysfunction, bladder and bowel incontinence, tingling, and numbness, as well as vision impairment, are some of the several neurological symptoms that might occur in MS, and according to their severity and evolution, the disease can take a variety of clinical courses. Relapsing–remitting (RR) MS, which affects 80% of patients, is characterised by attacks accompanied by full or partial remissions [4]. The term “progressive MS” refers to a subtle decline in neurological function accompanied by new symptoms and signs that have lasted at least a year. The main cause is neurodegeneration, which is manifested by extensive neuroaxonal degeneration in the white and grey matter [5]. A combination of clinical and radiographic (magnetic resonance) characteristics are used to diagnose MS; in detail, the guidelines for the recognition of all forms of MS are the McDonald criteria from 2017 [6]. The interaction of genetic human leukocyte antigen (HLA) DRB1, vitamin D deficiency, Epstein Barr Virus (EBV) infection, and epigenetic variables is a key component of the pathogenesis of MS [7,8,9,10]. The primary proposed aetiology of MS is an autoimmune attack on the central nervous system (CNS); in fact, studies have suggested that autoreactive T cells, mainly T helper (Th)-1 CD4+ T cells and Th17 cells, play a key role in the pathogenesis, especially since they secrete cytokines and trigger the inflammatory cascade [11]. Microglia activation and prolonged oxidative stress caused by the inflammatory demyelination process contribute to neurodegeneration and subsequently axonal and neuronal destruction [12]. Both inflammation and demyelination, as well as astroglial growth (gliosis) and neurodegeneration, define MS. Scientists identified metabolites differently active in MS during the four seasons, establishing the role of the citrate cycle, sphingolipid metabolism, and pyruvate metabolism. The fall and spring seasons were shown to have the most metabolites influenced by these changes, while summer had the least amount of metabolite changes. Histidine and its metabolite, methyl histidine, serum level seem to be lower in MS than in control: histidine is a precursor to histamine, a potent neurotransmitter and immune-modulator; the inverse relationship between the blood level of this amino acid and IL-6, a cytokine that promotes inflammation, may help to explain the immune-modulating role of histidine in the pathophysiology of MS. Similar to this, reduced levels of histidine were linked to elevated levels of oxidative stress indicators and serum C-reactive proteins. According to the scientific literature, low levels of histidine during the spring and fall may diminish the anti-inflammatory effects of this amino acid and its metabolites, which may lead to MS flare-ups. It should be mentioned that fall and spring when sunshine exposure significantly varies, cause fluctuations in histidine levels in MS, even if the specific mechanisms underlying these modifications in histidine levels are still unknown. Ceramides are lipid molecules that build up on the soma and axon of neurons, contributing to neuronal adhesion, ion channel regulation, and the expression of neurotransmitter receptors; on the other hand, oxidative stress brought on by an elevated ceramide level may cause neuronal apoptosis and death. The sphingomyelin–ceramide–sphingosine-sphingosine-1 phosphate (S1P) pathway produces ceramides and sphingosine, which are elevated in MS serum. Furthermore, ceramides play a role in the pathogenesis of MS; in fact, elevated serum levels of ceramides are one recurring alteration in MS metabolites. Moreover, research focused on the function of vitamin D and its derivatives in the etiology of MS, and it was discovered that the only season in which the amount of vitD was not higher than in control was fall: vit D serum levels were shown to be lower in the winter, spring, and summer. Globally, it can be stated which more overlapping metabolites were impacted in MS during the spring and fall seasons, and a return of symptoms during these two seasons could be explained by this fact [13].

## 2. Non-Coding RNAs

Although most of the human genome is actively involved in genetic transcription processes, eukaryotic cells are quantitatively poor in coding RNA (less than 1.4%), resulting in a large part of human DNA being transcribed into ncRNAs [14]. This concept partially overturns the biological dogma proposed by Francis Crick, according to which proteins represent the main and final product of genetic information [15]. On the other hand, ncRNAs, although not involved in ribosomal proteosynthesis, play a key role in many physiological and pathological cellular processes [16]. The optimization and innovation of gene profiling techniques have made it possible to isolate and sequence an ever-increasing number of ncRNA “species”. Gene sequencing methods based on next-generation sequencing (NGS), such as RNA-Seq or small RNA-sequencing (smRNA-Seq), ideal for identifying small RNA species using reverse transcription and PCR amplification, have implemented our knowledge about RNA, allowing the drafting of dedicated genetic databases [17]. The consultation of these “genetic libraries”, such as GENCODE, the RefSeq project, NONCODE, HUGO, LNCipedia, and the ENCODE project, represents an obligatory step to define the taxonomy of ncRNAs [14,18,19,20,21,22]. One of the parameters used to classify ncRNAs concerns the length and structure of the non-coding segment [23,24]. We can, therefore, distinguish the “small-ncRNAs”, composed of a few nucleotides, and the “long-ncRNAs” (lncRNA), whose length exceeds 200 nucleotides [25]. Small-ncRNAs include micro-RNAs (miRNAs, 22 nucleotides) and PIWI-interacting RNAs (piRNAs, 24–30 nucleotides). Circular-RNAs (circ-RNAs) are also grouped within the class of lncRNAs, although these have a circular structure unlike the other linear lncRNAs [25]. The ncRNAs can also be classified according to the function they cover within the eukaryotic cell, differentiating them into “Housekeeping ncRNAs” and “Regulatory ncRNAs” [26,27]. Housekeeping ncRNAs are generally small in size and perform basic cellular functions such as protein synthesis (tRNAs,rRNAs), RNA splicing (snRNAs), and post-transcriptional modification of rRNA (sno-RNA) [28,29]. Furthermore, among the Housekeeping ncRNAs, we include the tRNA-derived fragments, cleavage products of tRNAs, involved in the regulation of gene expression and apoptosis [30]. Regulatory ncRNAs perform functions of control and modulation of gene expression, both at the epigenetic and transcriptional levels [31,32]. The length of regulatory ncRNAs varies from a few bases (<200), as in miRNAs, siRNAs, and piRNAs, up to larger nucleotide sequences (>200), as in lncRNAs and circRNAs [33]. miRNAs are small portions of non-coding RNA, highly expressed within eukaryotic cells, which, through the formation of a biological aggregate called the “RNA-induced silencing complex (RISC)”, modulate the post-transcriptional processes of the mRNA, silencing specific target sequences [34,35]. Similarly to miRNAs, siRNAs also perform an RNA interference function through the activation of the RISC; PiRNAs, short non-coding elements composed of single-stranded RNA nucleotide sequences, form cytoplasmic complexes with PIWI proteins and migrating into the cell nucleus, actively modulate gene transcription [36]. 

### 2.1. Circular RNAs

Circular RNAs (circRNAs) are nuclear biomolecules composed of ribonucleotide sequences with a circular structure and no polyadenylated tails [37]. They are classified among the Housekeeping ncRNAs, although some forms possess the ability to code for specific proteins. A total of 1976 small portions of non-coding material with a circular structure, considered splicing errors, were isolated from viroids infecting plants and, subsequently, identified in the hepatotropic virus δ and eukaryotic cells, as in the mitochondrial RNAs of some yeasts [38,39,40,41]. In the 1990s, the casual discovery of circular and non-polyadenylated RNAs derived from non-canonical splicing of the human EST-1 gene and the isolation of the same molecules in the cytoplasm of mouse testicular cells aroused the interest of the scientific community towards this biotype of RNAs [42,43,44].

#### 2.1.1. circRNAs Identification Tools

There are many different forms of circRNAs with great stability in humans, according to studies performed with second-generation sequencing techniques and bioinformatics [45]. CircRNA identification and quantification require specially created bioinformatic pipelines due to the unique properties of the circRNA junction. In detail, a circRNA can be recognized by its back-spliced junction (BSJ) read, which serves as a molecular signature and connects two related exons in the reverse order of how they appear on the reference sequence [46]. Novel techniques for circRNA validation and identification are described in recent studies. One of these is the examination of non-polyadenylated libraries that had been pre-treated with RNase R to enrich them for circular RNAs; using this technique, it is possible to distinguish between genuine circRNAs and mRNAs with scrambled exons and to enrich for circRNAs [47]. Importantly, the existence of scrambled junctions can also result from frequent reverse transcriptase errors and other non-canonical splicing processes (such as trans-splicing). Integration of various circRNA identification tools has been shown to lower the false-positive rate [48].

#### 2.1.2. CircRNA Biogenesis

In 2012, Salzman and colleagues first proposed that pre-mRNAs can create circRNAs through reverse splicing, denominated “back-splicing” [49]. The spliceosomal machinery serves as the catalyst for the classic eukaryotic pre-mRNA splicing process, which involves cutting off introns and joining exons. Canonical splicing produces a linear RNA transcript with a 5′ to 3′ polarity in conjunction with other co-/post-transcriptional processing steps, including 5′ capping and 3′ polyadenylation. In contrast to canonical splicing, back-splicing (Figure 1) entails attaching an upstream (3′) splice acceptor site in the reverse manner to a downstream (5′) splice donor site. This results in the production of an alternatively spliced linear RNA with skipped exons and a covalently closed circRNA transcript [50]. Most circRNAs have been shown to come from exons in the gene’s coding region, but some have also been observed to come from non-coding regions as introns, intergenic areas, 5′- or 3′-UTRs, antisense RNAs, and 5′- or 3′-UTRs [51]. Exon-derived circular RNAs (EcircRNAs) are mostly found in the cytoplasm and originate from a back-splicing mechanism [52]. Circular intronic RNAs (ciRNAs) are a different class of circular RNA molecules originating from the lariat introns of Pol II transcripts [53] and have limited enrichment for miRNA target sites as opposed to accumulating primarily in the nucleus to regulate gene transcription in cis by boosting Pol II transcription of their parental genes through unknown processes [26,54]. The splicing factors, especially spliceosomes and cis-regulatory elements, considerably facilitate back-splicing, which takes place during the creation of circRNA [55]. Drosophila’s studies show that inhibition or silencing of spliceosomal components significantly boosted the production of circRNAs while reducing the amount of linear mRNAs [56]. Circularization has been modelled by three different processes, intron pairing-driven, RNA-binding protein (RBP)-mediated, and lariat-driven (ciRNAs), but the precise mechanism by which circRNA is produced is still unknown [57]. Intronic complementary sequences (ICS) are necessary for circularization by back-splicing, which facilitates the generation of circRNA by placing the donor and acceptor splice sites close together [58]. For example, removing ICS elements from the introns surrounding the general control of amino-acid synthesis 1-like 1 (GCN1L1) exon decreased the level of circGCN1L1 [59]. RBPs are essential for fostering the creation of tissue-specific circRNAs and can participate in circRNA production by binding certain motifs in adjacent intron sequences: for instance, DEAH-box helicase 9 (DHX9), which contains both a dsRBD and an RNA helicase domain, regulates circRNA synthesis by suppressing intron pairing brought on by ALU repeats. On the other hand, DHX9 removal raises the amount of circRNA [60].

#### 2.1.3. circRNAs Functions

CircRNA was originally thought to be a splicing by-product, making it unimportant and accidental in terms of biology. Integrating investigations have started to show that at least some circRNAs have potential significance in both healthy and pathological situations [61]. The majority of circRNAs’ functions are still unknown, but those that have been investigated appear to be predominantly involved in controlling gene expression, either directly or, more frequently, indirectly, by controlling other factors that control gene expression, such as miRNAs or RBPs. CircRNAs can act as miRNA sponges to control the production of miRNAs since they include miRNA binding sites [62]. For example, CircHIPK2 may operate as a sponge for miR124-2HG to regulate astrocyte activation by coordinating with autophagy and endoplasmic reticulum stress [63]. In addition, CircRNAs have the ability to act as scaffolds and protein sponges or transfer proteins from the cytoplasm to the nucleus: for instance, CircFoxo3 is particularly prevalent in the mammalian heart, and when it binds ID-1, E2F1, HIF1, and FAK, accelerates cardiac senescence through the retention of these proteins in the cytoplasm [64]. The circANRIL, which is linked to atherosclerotic cardiovascular disease, is another example: it binds to the vital 60S-preribosomal assembly factor pescadillo homolog 1 (PES1) and inhibits ribosome synthesis in macrophages and vascular smooth muscle cells, leading to nucleolar stress and cell death, which are important biological processes in atherosclerosis [65]. Furthermore, circRNAs can function as protein scaffolds to promote protein–protein interactions. A combination between CircFoxo3, p53, and MDM2 facilitates MDM2-induced p53 ubiquitination, which enhances p53 degradation [66]. They may possibly play a role in controlling transcription: recently, it was found that an insulin-derived circRNA interacts with RBP TDP-43 and is essential for regulating the transcription of genes involved in insulin production [67]. CircRNAs have the ability to produce proteins through translation in addition to controlling transcription. It has been shown that human cells have a large number of m6A motifs on their circRNA and that just one m6A site is required to start the translation of circRNA with the involvement of several proteins [68]. They can be translated by the m6A alteration, as well as in vivo or in vitro, when it has an internal ribosome entry site (IRES) [69]. Additionally, circRNAs are distinguished from other forms of RNA by extraordinarily durable stability and tissue-specific expression, making them ideal candidates for biomarkers [70]. 

## 3. Multiple Sclerosis and circRNAs

### 3.1. Epigenetic Mechanisms

DNA methylation is a potential regulatory mechanism that might modulate the production of circRNA. Recent studies in genetics and epigenetics have indicated that a significant proportion of potential causative variations for autoimmune disorders are located in non-coding regions of the genome [71]. These variants are believed to have a regulatory function in the expression of genes associated with these illnesses. Epigenetic processes have the ability to govern the expression of genes through the alteration of DNA, a process that may be inherited across successive cell divisions [72]. The epigenetic mechanism that has been extensively researched is the covalent attachment of a methyl group to cytosines within the CpG dinucleotide context. This particular process has garnered significant attention because of the well-established and steady process by which mCpG is propagated across DNA, facilitated by DNA (Cytosine-5)-Methyltransferase 1 [73]. DNA methylation changes have been seen in blood, CD4+, and CD8+ T lymphocytes, as well as in unaffected brain areas of individuals with MS [74]. Numerous research has been conducted to elucidate the role of epigenetic processes in the modulation of circRNA production, particularly within the domain of cancer. Evidence indicates that hypermethylation of cancer-specific promoter CpG islands is linked to reduced production of both circRNAs and their corresponding host gene counterparts [75]; additionally, it is well-established that intragenic methylation plays a role in the regulation of AS [76]. Furthermore, another work demonstrated that there were differential expressions of circRNAs with distinct methylated sites in tumour samples compared to adjacent normal samples; interestingly, this differential expression was not observed in the parental genes of these circRNAs: this finding suggests the possibility that certain abnormal DNA methylation events may specifically impact the processing of pre-mRNA to produce circRNAs, while not affecting the generation of linear RNAs [77]. In a recent study, an evaluation to determine the potential influence of DNA methylation on the regulation of back-splicing in the Jurkat cell line (immortal T cell) was conducted and provided evidence supporting the presence of a positive association between circRNA expression and gene body methylation (Spearman ranking analysis). It can be stated that epigenetic characteristics may have a significant impact on the composition of the cellular circRNA reservoir [78]. In the aforementioned study, in order to obtain a more comprehensive understanding of the influence of methylation on the back-splicing profile, a clustering approach was employed to categorise circRNAs into two groups based on their methylation beta values: a “low” methylation-level group and a “high” methylation-level group. The study encompassed both promoter and gene body methylation, with the assignment to a certain group determined by comparing the methylation level to the median threshold. Regarding gene body methylation, tests revealed a noteworthy disparity in circRNA expression between the groups categorised as having “low” and “high” methylation: specifically, a greater amount of methylation was shown to be correlated with elevated production of circRNA [78,79]. Moreover, a correlation was seen between the DNA methylation gene data in individuals with MS compared to healthy individuals and the identification of 36 genes with dysregulated circRNA profiles using RNA sequencing [80]. These data provide more evidence to justify the exploration of the relationship between circRNA and methylation profiles; by investigating this link, researchers may obtain a deeper understanding of the regulatory characteristics that influence the circRNA panorama and eventually shed light on the mechanisms behind disease pathophysiology [78].

### 3.2. circRNAs Biomarkers in MS

The exceptional stability of circRNAs, attributed to their resistance to exonucleases responsible for degrading linear transcripts, makes these non-coding molecules ideal as potential disease biomarkers. Additionally, circRNAs have expression patterns that are specific to different tissues and developmental stages, which may be a good way to address the lack of specificity seen in a number of existing biomarkers [80]. It is important to note that circRNAs exhibit stronger interspecies conservation, which makes it easier to transfer biomarkers from animal models to people [81], and this fact suggests that GSDMD circRNA may have the ability to serve as an important biomarker for MS [82]. In rare instances, the existence of a pathological disease may be indicated by a reduced or entirely absent marker. In the scientific literature, an example of a particularly significant dysregulated circular RNA, hsa_circ_0007990, is presented to its particular relevance due to the association with the PGAP3 host gene, which encodes a protein known to play a significant role in regulating autoimmune reactions; it is responsible for the modification of fatty acids in glycosylphosphatidylinositol (GPI)-anchored proteins [81]. Notably, when PGAP3 is absent in mice, there is an increased T cell response and a worsened experimental immune-mediated encephalomyelitis phenotype, along with various symptoms resembling autoimmune disorders [82]. A recent study confirmed the finding of PGAP3 circRNA downregulation in MS patients, representing not only a possible disease biomarker but also being considered, in the future, as a “disease progression biomarker” [78]. Another noteworthy example involves the annexin A2 gene (ANXA2) [83]. Annexin A2 (ANXA2), a member of the annexin family, is a calcium-dependent phospholipid-binding protein with a molecular weight of 36 kilodaltons, mostly found on the surface of the majority of eukaryotic cells. It plays a significant role in several biological processes, such as exerting anti-inflammatory effects, facilitating Ca2+-dependent exocytosis, regulating immunological responses, and controlling phospholipase A2 activity [84,85]. Furthermore, the aforementioned protein is implicated in immune-mediated conditions like arthritis and antiphospholipid syndrome [78,86]. The findings of a study conducted in 2018 indicate a downregulation of circRNA profiles, specifically ANXA2 circRNAs (circ_0005402 and circ_0035560), in the bloodstream of individuals diagnosed with MS [87]. Based on existing evidence, a definitive correlation between the ANXA2 gene and MS has not been demonstrated, even if it has been shown to be involved in facilitating the traversal of the blood–brain barrier, a protective barrier that safeguards the central nervous system [88]. In addition, ANXA2 has been identified as a target of miR-155, an important microRNA involved in neuroinflammation at the blood–brain barrier [89] and crucial in the development of Th1 and TH17 cells, as well as the polarisation of myeloid cells in the context of MS. The level of expression of miR-155 has been seen to be considerably elevated in both PBMCs and active lesions, and it is positively associated with disease severity in people with MS as well as in animals with experimental autoimmune encephalomyelitis (EAE) [90]. This observation correlates with the downregulation of ANXA2, indicating a potential intricate interplay of miRNA, mRNA, and circRNA. Additionally, ANXA2 plays a significant role in the post-transcriptional control of gene expression and the transportation of miRNA and vesicles [91]; in fact, it controls the loading of miRNAs into extracellular vesicles (EVs) through a process that is reliant on calcium [91]. These mechanisms have been suggested as potential indicators for comprehending the intricate network of transcriptome control in the context of MS. Hence, the expression of ANXA2 circRNAs may serve as biomarkers for RR-MS, exhibiting favourable levels of specificity and sensitivity [87]. On the other hand, several investigations have revealed an elevated production of circRNAs in individuals with MS compared to healthy individuals [92]. Prospective biomarkers for MS were found as circRNAs derived from the genomic regions of the RELL1 gene (chr4:37633006-37640126), RNF149 gene (chr2:101898320-101911643), and CHD9 gene (chr16:53288349-53308214). Evidence of the overproduction of circRNAs in MS patients not only suggests the potential utility of circRNAs as disease biomarkers but also indicates a distinct modification in the transcriptome of patients suffering from this disease [87]. The observed increase is unlikely to be associated with any changes in the spliceosomal machinery (cis-regulatory elements; RBPs) or alterations in the fraction of leukocyte cells studied. It is plausible that the upregulation of circRNAs is indicative of their involvement in the pathogenesis of MS [92]. 

### 3.3. Extracellular Vesicles and circRNAs

Extracellular vesicles (EVs) refer to entities that are covered with a membrane and originate from either endosomes or the plasma membrane, then released into the extracellular environment [93]. The EVs were initially discovered four decades ago when they were observed in the form of reticulocytes; subsequently, it has been demonstrated that EVs may be found in many bodily fluids [94]. Typically, researchers have identified three distinct categories of extracellular vesicles (EVs): exosomes, which have a diameter ranging from 40 to 100 nm, microvesicles, and apoptotic bodies, with a diameter ranging from 50 to 2000 nm [95]. EVs are of paramount importance in facilitating intercellular relationships through the transfer of lipids, proteins, and genetic material between cells; in fact, their involvement in extracellular vesicle (EV)-mediated communication has been demonstrated in the modulation of multiple biological processes (Figure 2), such as the immune system’s response [96,97]. The RNA payload within EVs exhibits a great degree of complexity, with just a few research having comprehensively characterised the whole EV transcriptome in the context of MS [98,99]. In recent times, it has been demonstrated that circRNAs exhibit a high concentration within EVs and that exosomal circRNAs possess the ability to act as miRNA sponges, suppress protein activity, regulate splicing and transcription processes, and interact with RBPs [50,100]. These molecular functions have been found to have a role in the development and progression of neurodegenerative illnesses, although to a certain extent [101]. Regarding multiple sclerosis, Iparraguirre and colleagues revealed that circRNAs constitute the second most prevalent transcript in EV samples obtained from both MS patients and healthy individuals, showing disparities in the RNA biotypes present in leukocytes and EVs [102]. Interestingly, circRNAs have a higher representation in EVs, accounting for 4.2% of the total reads and 58.4% of the reads associated with ncRNAs, in contrast to leukocytes, where they constitute just 0.2% of the total reads and 0.9% of the ncRNA reads. Various parameters have been documented to influence the incorporation of RNA into EVs. These factors include cell abundance, particular sequence motifs, secondary structure, length, differential affinity for membrane lipids, and connection with RBPs [103]. In addition to the inherent characteristics of circRNAs that may facilitate or impede their inclusion inside EVs, the cellular state also influences the profiles of EV RNA. Consequently, a specific RNA molecule may be selectively encapsulated or excluded from EVs based on its physiologic or pathological consequences [102]. It is noteworthy to mention circNEIL3, one of the circRNAs identified as being differentially expressed in EVs of patients with RR-MS compared to EVs of healthy controls, and for this reason, has been proposed as a potential biomarker in leukocytes of RR-MS patients [92]. Conversely, a minimal fraction of the circRNA pattern in leukocytes exhibited variations between RR-MS and SP-MS patients. As a result, no possible biomarkers indicative of disease progression were identified. In conclusion, the biocompatibility of EVs, specifically of exosomes, and the specific functionality of ncRNAs, an increasing number of researchers have been investigating the potential of using exosomal ncRNAs as a viable therapeutic approach for central nervous system disease [104,105].

### 3.4. circRNAs Genetic Variation in MS

Paraboschi and his group conducted a bioinformatic investigation to examine the enrichment of circRNAs originating from non-coding regions in the genome related to MS and proposed that these circRNAs may have a role in the susceptibility to the illness. The authors of the study have shown a significant increase in non-coding elements, particularly circRNAs, that are located inside the 73 Linkage Disequilibrium (LD) blocks containing single-nucleotide polymorphisms (SNPs) linked with MS: in detail, a collective count of 482 circRNAs was identified in publically accessible databases and then compared to an average of 194.65 circRNAs found in randomly selected sets of LD blocks, using 1000 iterations. Through the assessment of RNA sequencing data obtained from two distinct cell lines, namely SH-SY5Y and Jurkat cells, a total of 18 circRNAs were discovered: among them, two were found to be unique and originated from genes related to MS. Furthermore, Paraboschi conducted an investigation to validate the levels of expression of a circRNA originating from a genomic region linked with MS, namely hsa_circ_0043813 originated from the STAT3 (Signal Transducer and Activator of Transcription 3) gene [101]: the activation of STAT3 is mediated by a diverse range of cytokines, which in turn, elicits a multitude of crucial biological activities and it has recently shown to be involved in the control of Th17 cell development, which is known to be a critical factor in the pathogenesis of MS [102]. Scientists examined the relationship between STAT3 hsa_circ_0043813 and the expression levels of certain circRNAs, which may be altered by disease-associated SNPs [106,107]. This finding aligns with the previously reported evidence on circANRIL, which stands as the sole instance of a circRNA where a connection between disease-associated SNPs and circRNA formation has been established [108]. In conclusion, researchers found unique evidence indicating that the top hits from genome-wide association studies (GWAS) in MS are located inside LD blocks that are enriched in circRNAs, suggesting that they may potentially play a role in the pathogenesis of MS, representing a new avenue for further investigation [107].

### 3.5. circRNAs and B-Cell Function

B-cell function plays a crucial role in the autoimmune aberrations observed in RRMS. In recent years, there has been substantial evidence supporting the significant involvement of B lymphocytes in several processes inside the CNS. These processes include antigen presentation, the release of cytokines that induce inflammation, and the generation of antimyelin antibodies [109,110]. There is also a proposition suggesting that ectopic lymphoid cell aggregation found in the leptomeninges consists of B lymphocytes and could have a role in determining the chronic nature of MS illness disease [111]. Recent studies have provided confirmation that the reduction of B-cells is an effective treatment strategy for treating RRMS and primary progressive multiple sclerosis (PPMS) and specific circRNA particles (hsa_circRNA_101348, hsa_circRNA_102611, and hsa_circRNA_104361) that are overproduced in PBMCs of patients with RRMS have been identified [112,113]. Importantly, the authors of this study have discovered two transcripts, AK2 and IKZF3 (Aiolos), which are associated with B-cell function and are regulated by these circRNA molecules. The results were validated by the direct demonstration of an increase in AK2 and IKZF3 expression in the peripheral blood mononuclear cells (PBMCs) of individuals experiencing a relapse of MS. The AK2 gene is recognised for its role in encoding phosphotransferase adenylate kinase 2, which fulfils the unique cellular needs related to mitochondrial processes, notably in the context of B-cell activation and antibody generation [114]. A special emphasis on the involvement of AK2 has been observed in B cells that are infected with the Epstein–Barr virus (EBV). This association has been extensively explored in the context of MS as a viral infection that is linked to MS [115]. The IKZF3 gene encodes the proteins Aiolos and Ikaros, which serve as crucial transcription elements belonging to the Ikaros family of zinc-finger proteins. These proteins play a pivotal role in regulating the development of lymphoid and myeloid cells, as well as maintaining immunological homeostasis [116]. It is noteworthy that a mutation of the IKZF3 gene has been identified as one of the risk alleles associated with MS [117]. The involvement of Aiolos in the functioning of mature B cells has been demonstrated, highlighting its crucial importance. Moreover, Aiolos is essential for the development of high-affinity antibody-secreting plasma cells [118]. Previous research has indicated that Aiolos plays a crucial function in the latter phases of B-cell development [119]. Therefore, the manipulation of Aiolos expression by circRNA has the potential to play a role in the aberrant immunological response mediated by B-cells in multiple sclerosis (MS). More specifically, it might potentially induce B-cell differentiation towards the production of autoantibodies. Based on the aforementioned results, it is justifiable to conduct more studies pertaining to B-cell populations. Another transcript, CBX5, which was inferred from the analysis of differentially expressed circRNA/miRNA interactions in the work conducted by Zurawska et al. [114], has been associated with several biological processes such as stem cell self-regeneration, lineage commitment, as well as carcinogenesis and maturation [120]. In conclusion, the circRNAs have a distinct expression pattern characterised by their interaction with miRNA through a sponge-like mechanism, resulting in the formation of a circRNA/miRNA network that is distinctive to RRMS. The presented data exhibit potential for the advancement of novel biomarkers in the context of RRMS. Furthermore, Zurawska et al. findings propose an unexplored function of circRNAs in modulating the transcriptional programme of B cells in the context of [114].

### 3.6. Roles of circRNAs in MS

#### 3.6.1. hsa_circ_0106803 of GSDMB Gene

The available evidence suggests that circRNAs play important roles in controlling how the immune system and central nervous system (CNS) function. Only a few research have looked at this issue; thus, the precise roles played by these RNAs in the onset of MS are yet unknown. In contrast to PBMCs from healthy people, patients with RRMS showed differential expression of approximately 400 circRNAs in their peripheral blood mononuclear cells (PBMCs), according to the research of Cardamone and colleagues about “alternative splicing” (AS) and, specifically, the detection and description of GSDMB, a 17q12-locus alternatively spliced gene that has been frequently linked to autoimmune disorders and asthma susceptibility and that encodes Gasdermin B, a member of the family of proteins that contain gasdermin domains [121,122,123]. Alternative splicing (AS) has become more and more significant in the pathogenesis of autoimmune diseases in recent years since numerous immune system processes, including T-cell activation and migration, cytokine response, and T-cell stability and apoptosis, are impacted by it [124]. These processes are all necessary to prevent the suppression of self-tolerance and the emergence of autoimmune conditions. Previous studies have identified a particular polymorphism of the GSDMB gene (denominated “rs11078928”) in autoimmune disorders, characterized by the synthesis of a transcript without exons 5, 6, 7, and 8 and also modifications to the exon 6 skipping rate, in particular, the pattern GSDMB AS in RR-MS patients, with levels of exon 6 and exons 5–8 skippings independent to the rs11078928 polymorphism [125,126,127]. For at least two additional genes, PRKCA and NFAT5, this similar dysregulation of AS profiles in MS patients has already been observed, representing a possible “mark” of this disease’s condition [128]. As previously stated, AS represents an important mechanism for the genesis of circRNAs. The GSDMB AS back-splicing pattern gives rise to an ecircRNA (including exons 5 and 4), which is annotated in both the circBase database (accession number hsa_circ_0106803) and the CIRCpedia database (accession number HSA_CIRCpedia_78516) and is expressed in several parts of the brain [83]. The analysis of GSDMB ecircRNA expression conducted by Cardamore et al. in the sample population revealed a significant increase of 2.8-fold in the expression levels of GSDMB circRNA in peripheral blood mononuclear cells (PBMCs) of patients with relapsing-remitting MS (RR-MS). The determination of the potential role of changes in circRNA synthesis and GSDMB alternative splicing (AS) in the development of susceptibility to multiple sclerosis (MS) continues to pose challenges [129]. In support of a potential involvement of GSDMB in the development of multiple sclerosis (MS), it was demonstrated that reducing the expression of GSDMB in memory CD4+ T-cells resulted in an increased production of cytokines, including tumour necrosis factor (TNF), interleukin (IL)-13, and IL-16 [130]. Furthermore, a study revealed that the overexpression of the GSDMB D6 isoform in primary human bronchial epithelial cells resulted in elevated expression levels of many genes, including transforming growth factor beta 1 (TGFB1) and matrix metallopeptidase 9 (MMP9) [131]. The study demonstrated a reduction in TGFB1 expression levels in leukocytes of individuals with multiple sclerosis (MS) [132]. Additionally, it was shown that TGFB1 expression levels were elevated in the serum of MS patients following interferon beta-1b (IFN-β1b) therapy [133]. The dysregulation of inflammasome signalling has been implicated in several autoimmune and inflammatory diseases, such as MS, and represents another possible pathogenetic mechanism of GSDMD [134]. It was demonstrated that pyroptosis, a kind of cell death that is dependent on the inflammasome, is favored by the action of a member of the Gasdermin family, known as GSDMD [135]. The activation of this pathway is initiated by the GSDMD N-domain, which is liberated by proteolytic cleavage mediated by inflammatory caspases [136]. Furthermore, it has been observed that the N-domains of the GSDMB protein possess the capacity to induce pyroptosis in HEK293T cells [137]. As a corollary to what has been stated above, the exceptional stability of circRNAs, attributed to their resistance to exonucleases responsible for degrading linear transcripts, along with their abundant expression in peripheral whole blood, suggests that GSDMD circRNA might serve as a promising biomarker for MS [138].

#### 3.6.2. circ_HECW2 and the Dysfunction of the Blood–Brain Barrier

Additionally, it was discovered that circular RNA HECW2 (circ_HECW2) contributed to the pathogenesis of multiple sclerosis. In both in vitro and in vivo MS experimental models, increased expression of circ_HECW2 results in EndoMT, which is crucial for BBB failure and contributes to BBB leakage. In order to boost the expression of ATG5 (autophagy-related 5), activate the NOTCH pathway, and ultimately favorably regulate LPS-induced EndoMT, Yang and colleagues demonstrated that circ_HECW2 served as a miR-30D sponge. Further research by Dong and his group demonstrated that circ_HECW2 also interacted with miR-30e-5p to control neural growth regulator 1 levels, which suppressed endothelial cell proliferation and increased apoptosis and LPS-induced EndoMT [125,126].

#### 3.6.3. hsa_circ_0106803 Modulates the Expression of ASIC1a mRNA

MS patients had been discovered to have an increase of the lncRNA MALAT1: changes in circRNA back-splicing and aberrant splicing of MS-related genes, including IL7R and SP140, have been linked to changes in MALAT1 expression [78]. MiR-1275 and miR-149, two miRNAs that were among those predicted by hsa_circ_0106803 to have multiple target sites, exhibit differential expression in blood from MS patients; moreover, ASIC1a is said to bind to miR-149, which lowers its levels. The acid-sensing ion channel subunit ASIC1a is overexpressed in acute MS lesions and may play a role in the neuronal pathophysiology of the illness: by controlling the expression of ASIC1a mRNA through miR-149, hsa_circ_0106803 has been suggested to affect the progression of MS in light of this data [127].

#### 3.6.4. circINPP4B Regulates Th17 Cell Differentiation

CD4+ T helper (Th) cells, particularly Th17 cells that secrete interleukin-17 (IL-17), play a significant role in the initiation and progression of both MS and experimental autoimmune encephalomyelitis (EAE), which is a mouse model used to study MS [139,140]. To support this concept, it was observed that there was a notable rise in the proportion of Th17 cells in the cerebrospinal fluid (CSF) of individuals diagnosed with MS. Furthermore, it was shown that the proportion of Th17 cells in MS patients during relapse was larger compared to the ratio observed during the remission phase [102]. The induction of EAE may be achieved by the infusion of activated CD4+ Th17 cells that are responsive to myelin. Additionally, the administration of IL-17 exacerbates the progression of EAE, whereas animals lacking IL-17 demonstrate resistance to this illness [141]. A recent investigation has demonstrated that the inhibition of Th17 differentiation in immune-deficient Rag1−/− mice who were administered CD4+ naïve T cells (CD4+ Tn) led to the reduction of the signs of EAE. In contrast, the promotion of Th17 differentiation worsened the EAE condition [139]. In 2016, a team of investigators made a significant discovery regarding the expression of miR-30a in peripheral blood CD4+ T lymphocytes of individuals with MS and mice EAE: this study revealed that there was a significant reduction in the expression of miR-30a in these subjects. Furthermore, researchers found that the overexpression of miR-30a had the ability to effectively inhibit the excessive differentiation of Th17 cells, which are known to play a role in the pathogenesis of MS and EAE. Additionally, this overexpression of miR-30a was found to reduce the inflammatory response in the demyelinated region of the central nervous system (CNS), thereby providing relief from EAE symptoms [142,143]. Han investigated the circRNAs that have a role in the course of EAE in mice, building upon the existing literature in this field, conducting an analysis of the circRNAs that were differently expressed and putting in evidence that the expression of circ_3998 was not only considerably elevated in mice with EAE, but also showed a progressive increase as the EAE scores rose. This indicates that circ_3998 may have significant implications in the evolution of EAE. Based on the microarray data and subsequent bioinformatics research, it was determined that circ_3998 is situated on chromosome 8. This circRNA is comprised of six exons, namely exons 3 to 8, originating from the INPP4B gene. The same study demonstrated a positive correlation between the severity of EAE and the upregulation of circINPP4B expression in CD4+ T cells. Furthermore, it was shown that circINPP4B exhibited significantly elevated levels of expression in CD4+IL-17+ T cells and modestly increased levels of expression in CD4+IFN-γ+ cells and CD4+CXCR5+ cells, as compared to CD4+ Tn cells. Furthermore, in the context of in vitro Th17 cell development, the inhibition of circINPP4B resulted in a decreased proportion of Rorγt, a gene marker associated with the Th17 lineage, as well as a reduction in the production of Th17-related cytokines. These findings suggest that circINPP4B has a role in the regulation of Th17 development in an in vitro setting and, by extension, in the development of EAE. To corroborate this hypothesis, the mice in which the circINPP4B gene was silenced exhibited a mitigated form of EAE, characterised by a delayed start and lowered peak scores of EAE severity. The localization of circINPP4B in the cytoplasm, as shown by the FISH experiment, suggests that circINPP4B has the potential to function as a miRNA sponge. In order to elucidate the relationship between circINPP4B and miR-30a, Han et al. observed a progressive upregulation of circINPP4B expression and a concurrent downregulation of miR-30a expression during in vitro Th17 cell development. The downregulation of circINPP4B in mice resulted in an elevation in the expression level of miR-30a. In order to assess the impact of the circINPP4B/miR-30a axis on Th17 cell development, the investigators conducted a rescue test including the overexpression of circINPP4B and miR-30a, along with mutant variants of circINPP4B and miR-30a. The findings of this study indicate that in the context of Th17 induction in vitro, miR-30a exerts inhibitory effects on Th17 cell production. Conversely, circINPP4B has a promotive role in Th17 cell differentiation by enhancing the amount of IL-17+ cells, secretion of Th17-related inflammatory mediators, and expression of Rorγt and IL-21R. Furthermore, it was shown that the inhibitory effects of miR-30a were somewhat counteracted by circINPP4B. In summary, there is a correlation between the expression of circINPP4B and the clinical prognosis of MS. In their study, Han et al. investigated the expression level of circINPP4B in peripheral blood lymphocytes derived from a cohort of 18 patients diagnosed with RR-MS. The test results revealed a considerable upregulation of circINPP4B expression and downregulation of miR-30a expression in MS patients during the stage of relapse, as compared to healthy subjects who were matched in terms of age and sex. Furthermore, it was shown that the expression levels of circINPP4B and miR-30a in patients with MS recovered to normal while they were in a state of remission. This contrasted significantly with the expression levels observed in individuals experiencing a relapse. The results of this study suggest a potential association between the level of circINPP4B and the development of RR-MS. Hence, this research contributes to a more comprehensive comprehension of functional circular RNAs (circRNAs) in the context of multiple sclerosis (MS) and proposes a prospective diagnostic and therapeutic target for the management of Th17-mediated MS pathology [48].

#### 3.6.5. hsa_circRNA_101145 and hsa_circRNA_001896 in Patients with Relapse-Remitting MS

A complex analysis of the RNA of the peripheral blood mononuclear cells from patients with multiple sclerosis demonstrated several hundred circRNAs differentially expressed respect healthy controls (HCs). Mycko and colleagues evaluated 102 Caucasian participants, comprising 37 HCs and 65 patients with multiple sclerosis. All patients had relapsing-remitting multiple sclerosis, fulfilled the McDonald criteria, had not received disease-modifying drugs from any less than 6 months, and never had anti-CD20, cladribine, and anti-CD52 treatment. In order to distinguish RRMS patients in remission from HCs, this study identified two circRNAs and a collection of 10 miRNAs that may be targeted by circRNAs. All 10 of these miRNAs would be more active if circRNA were to express itself less. Numerous miRNAs have been connected to autoimmune mechanisms causing demyelination in MS. MiRNA has been linked by researchers to the growth and upkeep of populations of T helper cells that are encephalitogenic. MiRNAs are indeed variably expressed in PBMCs, whole blood, T cells, and B cells in MS patients, according to reports on the subject. The greater concentration of miR-181c in the cerebral fluid has been associated with a higher risk for the development of confirmed MS from the group of miRNAs discovered in this investigation. Almost all of the other miRNAs have been discovered to be involved in processes like autophagy, apoptosis, neural stem cell proliferation, and neurodegeneration, which may function in autoimmunity, brain recovery, and MS-related processes even if they have not been explicitly linked to the disease. The results gathered here provide a solid MS-related backdrop for the actions of the two highlighted circRNAs. The fact that NOS1, NOPCHAP1, and DSE, three protein-coding genes regulated by both circRNA molecules found in this study, have been described by in silico analysis is very pertinent. The blood–brain barrier’s permeability, microglia activity, and oxidative stress in immune cells and tissue in MS have all been linked to NOS1. In their respective biosynthetic processes, chondroitin sulfate, dermatan sulfate, and heparan sulfate covalently bond to particular core proteins to create proteoglycans. It is particularly intriguing because, in MS, chondroitin sulfate can significantly affect tissue remodeling and prevent remyelination. When extracellular matrix components like chondroitin sulfate and others are deposited into lesions, the resulting changed milieu prevents oligodendrocyte progenitor cells from proliferating. In order to prevent harmful neuroinflammation and to encourage the recruitment and maturation of oligodendrocytes in order to improve remyelination, a new prospective treatment strategy has recently been launched. The elevated expression of miRNAs that results from decreased circRNA expression as potential differentiating biomarkers for MS is another feature of our findings that may encourage further thought. In conclusion, they have shown that hsa_circRNA_101145 and hsa_circRNA_001896 are downregulated in the PBMCs of RRMS patients who are in remission. A circRNA–miRNA network primarily impacting post-transcriptional regulation is created by the expression of circRNAs that interact with miRNA regulatory mechanisms. The change in this network may offer microenvironmental modifications that control MS development [144].

## 4. Conclusions

The structural attributes of circRNAs, their distribution across the CNS, and their involvement in regulating the immune response render them promising candidates as “non-invasive” biomarkers for disease detection in individuals with MS, as well as prospective targets for novel treatment strategies [100]. Multiple studies have provided evidence of the widespread presence and differential expression of these molecules in individuals with MS in comparison to individuals without the disease. CircRNAs play a role in regulating the expression of cytokines, growth factors, and enzymes involved in the maintenance of neuroinflammation in patients with MS [145,146]. They can directly or indirectly influence gene expression, for example, by acting as miRNA sponges and regulating epigenetic mechanisms [63,147]. Additionally, circRNAs can modulate the activity of immune cells that are involved in the pathogenic mechanisms of MS (Figure 3). Moreover, the several aforementioned research emphasised the potential of these molecules to serve not only as disease markers but also as indicators of disease progression (Table 1). These findings suggest the possibility of developing diagnostic tools that may have the potential to alter the course of this illness in the future [45,148,149]. Hence, comprehensive investigations into the physiological and pathological functions of these non-coding molecules may potentially offer a novel diagnostic and therapeutic strategy for MS in the forthcoming years.

## Figures and Tables

**Figure 1 biomedicines-11-02883-f001:**
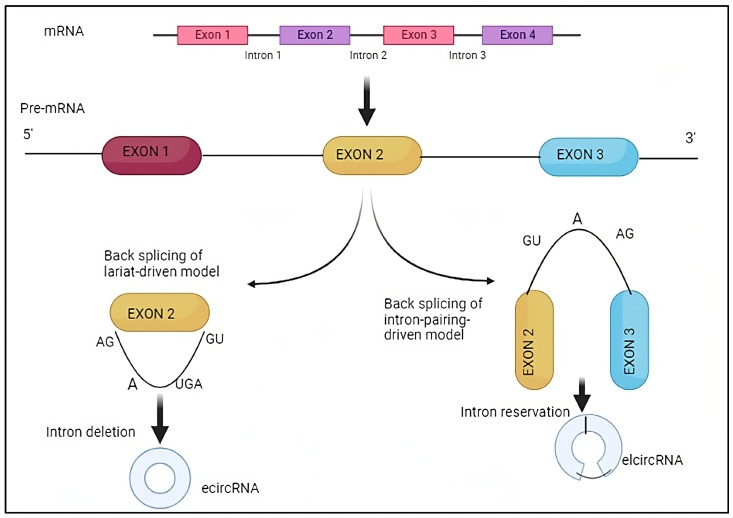
Back-splicing process: Back-splicing entails attaching an upstream splice acceptor site in a reverse manner to a downstream splice donor site, producing an alternatively spliced linear RNA with skipped exons and a covalently closed circRNA transcript. circRNAs have been shown to come from exons in the gene’s coding region and from non-coding regions as introns, intergenic areas, 5′- or 3′-UTRs, antisense RNAs, and 5′- or 3′-UTRs.

**Figure 2 biomedicines-11-02883-f002:**
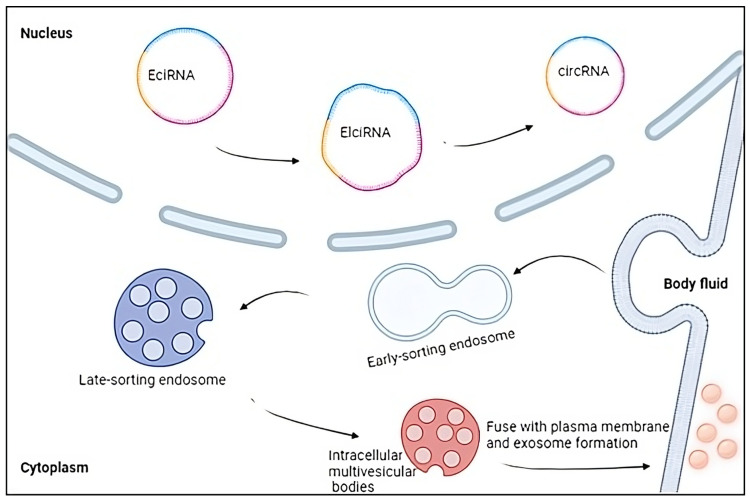
Extracellular vesicles and circRNAs: Exosomes are extracellular vesicles covered with a membrane and released into the extracellular environment. They modulate multiple biological processes, such as the immune system’s response, and contain a high concentration of circRNAs capable of acting as miRNA sponges, suppress protein activity, regulate splicing and transcription processes, and interact with RNA-binding proteins. Created with BioRender.com (accessed on 19 October 2023).

**Figure 3 biomedicines-11-02883-f003:**
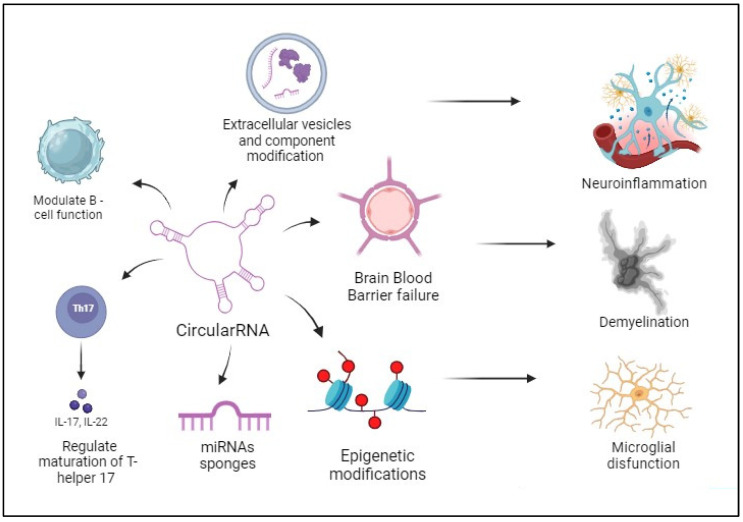
CircRNAs and pathogenetic mechanisms in MS: CircRNAs exert their effects through many pathogenic processes that promote the occurrence of characteristic alterations seen in MS, including neuroinflammation, demyelination, and microglial dysfunction.

**Table 1 biomedicines-11-02883-t001:** circRNAs in Multiple Sclerosis: circRNAs exhibiting differential expression in patients with Multiple Sclerosis.

circRNA	Gene	References
*hsa_circ_0106803*	GSDMB	Cardamone et al. [127]
*hsa_circ_0007990*	PGAP3	Cardamone et al. [75]
*circ_0005402* *circ_0035560*	ANXA2	Iparraguirre et al. [83]
*hsa_circ_0043813*	STAT3	Paraboschi et al. [101]
*hsa_circRNA_101348 hsa_circRNA_102611* *hsa_circRNA_104361*	AK2 IKZF3 CBX5	Zurawska et al. [114]
*circ_HECW2*	HECW2	Yang et al. [125] Dong et al. [126]
*hsa_circ_0106803*	ASIC1a	Cardamone et al. [127] Xia et al. [89]
*circ_3998*	INPP4B	Han et al. [35]
*hsa_circRNA_101145 hsa_circRNA_001896*	NOS1 NOPCHAP1 DSE	Mycko et al. [144]

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
