# Peer review of "Circular RNAs: A New Approach to Multiple Sclerosis"

_biomedicines, 2023, doi:10.3390/biomedicines11112883_

Round 1

Reviewer 1 Report

This paper offers a captivating exploration into the potential involvement of circular RNAs (circRNAs) in the pathogenesis of multiple sclerosis (MS). It injects a fresh and innovative perspective into the realm of MS research, illuminating a domain that holds substantial promise for advancing our comprehension of this intricate neuroimmune disorder. Here, I present some commendable attributes of the paper:

Figural Reference and Titles: It's essential to ensure that none of the figures remain unattributed in the paper. Properly referencing them within the respective sections and providing descriptive figure titles would enhance clarity.

Section 3 Comprehensive Overview: Section 3 delves into various aspects connecting MS and circRNAs. To facilitate improved understanding, it would be beneficial to encapsulate these multifaceted insights within 1-2 figures or tables.

In conclusion, this paper introduces an exciting perspective on the involvement of circRNAs in MS. It accentuates their potential to revolutionize the diagnosis and treatment of this formidable disorder. Its innovative approach, relevance to clinical practice, inclusion of recent findings, and potential to usher in a paradigm shift all contribute to its substantial value in the realm of MS research. With minor refinements in organization and enhanced clarity, this paper holds the potential to significantly impact our comprehension of multiple sclerosis and set the stage for future breakthroughs.

Minor editing of English language required

Author Response

Dear reviewer, I revise the paper accordingly to your suggestions.

Giuseppe Murdaca

Reviewer 2 Report

This manuscript explores the role of circular RNAs (circRNAs) in the pathogenesis of multiple sclerosis (MS), a condition characterized by immune-mediated demyelination and axonal damage in the central nervous system. The authors highlight the importance of non-coding RNAs in regulating immune system activity and gene expression, particularly in the context of autoimmune disorders like MS. The review discusses how circRNAs influence post-transcriptional control, microRNA expression, and epigenetic factors, contributing to the development of MS-related abnormalities such as neuroinflammation and damage to neuronal cells. The manuscript also suggests that circRNAs could potentially serve as non-invasive biomarkers for MS diagnosis and as targets for novel treatment strategies. Overall, it proposes that a deeper understanding of circRNAs in the context of MS may lead to new diagnostic and therapeutic approaches in the future. The manuscript can be published after addressing minor revisions.

The role of autoreactive lymphocytes in multiple sclerosis would benefit from a more recent reference (such as doi: 10.3389/fimmu.2022.996469) rather than the 2002 reference.

Additionally, the manuscript does not discuss seasonal changes in biomarkers in multiple sclerosis. For a recent publication discussing this issue, you can refer to https://doi.org/10.3390/ijms24043542

Author Response

(The authors gave the same response as above.)
